# Application of Nanomaterial Modified Aptamer-Based Electrochemical Sensor in Detection of Heavy Metal Ions

**DOI:** 10.3390/foods11101404

**Published:** 2022-05-12

**Authors:** Zanlin Chen, Miaojia Xie, Fengguang Zhao, Shuangyan Han

**Affiliations:** 1Guangdong Key Laboratory of Fermentation and Enzyme Engineering, School of Biology and Biological Engineering, South China University of Technology, Guangzhou 510006, China; 202121050086@mail.scut.edu.cn (Z.C.); 201921046785@mail.scut.edu.cn (M.X.); 2School of Light Industry and Engineering, South China University of Technology, Guangzhou 510006, China; fgzhao@scut.edu.cn

**Keywords:** heavy metal ions, aptamer, electrochemical biosensor, nanomaterials

## Abstract

Heavy metal pollution resulting from significant heavy metal waste discharge is increasingly serious. Traditional methods for the detection of heavy metal ions have high requirements on external conditions, so developing a sensitive, simple, and reproducible detection method is becoming an urgent need. The aptamer, as a new kind of artificial probe, has received more attention in recent years for its high sensitivity, easy acquisition, wide target range, and wide use in the detection of various harmful substances. The detection platform that an aptamer-based electrochemical biosensor (E-apt sensor) provides is a new approach for the detection of heavy metal ions. Nanomaterials are particularly important in the construction of E-apt sensors, as they can be used as aptamer carriers or sensitizers to stimulate or inhibit electrochemical signals, thus significantly improving the detection sensitivity. This review summarizes the application of different types of nanomaterials in E-apt sensors. The construction methods and research progress of the E-apt sensor based on different working principles are systematically introduced. Moreover, the advantages and challenges of the E-apt sensor in heavy metal ion detection are summarized.

## 1. Introduction

With the rapid development of industry and the improvement of urbanization, more and more chemical substances are used in daily life and agricultural production. Increasingly frequent industrial activities such as mining, metallurgy, and oil extraction produces many toxic and harmful substances. These toxic and harmful substances, even after purification treatment, will still leave some residues in the natural water system, including heavy metals, inorganic salts, and agricultural veterinary drugs, which cause pollution and damage the water environment [1,2,3]. Unlike organic pollutants, heavy metals cannot be biodegraded under natural conditions [4] and will be passively ingested by plants through drinking and irrigation, and eventually, will enter the human body through continuous accumulation in the food chain. Mercury, cadmium, lead, Chromium, Thallium, Antimony, and arsenic are the most common heavy metal pollutants. According to WHO standards, they usually do not exceed 2 ppb. The heavy metals ingested into the human body are likely to form complexes with biological substances such as proteins, enzymes, and nucleic acids. The formation of such complexes alters the molecular composition and mechanism of biological matter, causing it to fail to perform its original physiological function or causing distortion [5]. The accumulation of these elements can cause serious damage to the gut, bones, central nervous system, liver, kidneys, and reproductive system. Since these elements cannot be removed by normal removal methods, even trace amounts of heavy metals can pose a serious threat to living things [6].

In these cases, detection of heavy metal ions in environmental and water systems to prevent heavy metal pollution from the source of the food chain is a vital need. In recent years, many detection methods for heavy metal ions have been developed. Traditional detection methods mainly calculate the concentration of an atom based on its characteristic spectral intensity, including atomic absorption spectroscopy (AAS), inductively coupled plasma mass spectroscopy (ICP-MS), X-ray fluorescence spectrometry (XRF), neutron activation analysis (NAA), and inductively coupled plasma-atomic emission spectrometry (ICP-AES) [7]. These methods can perform accurate qualitative and quantitative analyses of heavy metal ions with high sensitivity, but they are also expensive and require laborious pre-processing [8]. Therefore, a cost-effective, fast, and efficient detection method for heavy metal ions needs to be developed.

Biosensors play an indispensable role in the development of biotechnology and are a fast analytical tool for detection at the molecular level. Usually, the recognition factor used in this technology is an antibody, but the emergence of the aptamer brings new prospects and possibilities to the field of biosensor analysis [9]. An aptamer is an artificial, single-stranded oligomer probe of DNA and RNA consisting of 10–100 bases, obtained by the SELEX index enrichment method. In Figure 1 the general process of SELEX is shown in detail. The initial ssDNA library was exposed in a container filled with target molecules, and the sequences with specific binding ability to the target were separated from the library. Then, a PCR reaction was used for bulk amplification and the next round of screening was carried out until ssDNA with the highest affinity to the target was obtained. Aptamers can fold into complex structures, selectively binding to the target with high affinity and specificity, known as artificial antibodies [10,11,12,13]. Compared with antibodies, aptamers have obvious advantages such as high chemical stability, low cost, easy operation, and they are easy to obtain [14]. Aptamers have a wide range of recognition targets, such as proteins, small molecules, agricultural veterinary drugs, bacteria, and heavy metal ions [15,16,17,18,19], which have broad application prospects in the field of food detection. A variety of biosensors using aptamers as recognition factors have been developed and applied to the detection of heavy metal ions, providing a new, efficient, and fast platform for the detection of heavy metal ions.

The electrochemical aptamer sensor (E-apt sensor), which is composed of biometric elements and signal sensors, has attracted more and more attention for this purpose. The signal sensor usually consists of an electrode substrate, modified layer, and electrochemical signal detection system. The most widely used electrode substrates include a gold electrode (AuE), glassy carbon electrode (GCE), indium tin oxide electrode (ITO), reduced graphene electrode (ERGO), and screen-printed electrode (SPE). Different electrode materials can have different degrees of signal enhancement after being modified by an appropriately modified layer and a variety of nanomaterials can be selected for the modified layer. The aptamer is fixed on the surface of the electrode by intermolecular force with the modified layer. This operation will change the impedance of the electrode and cause a current change. Therefore, nanomaterial modification in electrodes becomes an important part of electrochemical sensor construction. The electrochemical signal detection system includes a signal amplifier, a processor, and a display screen. The electrochemical sensor has been widely used in recent years because of its advantages of simplicity, high efficiency, strong specificity, and high sensitivity. As shown in Figure 2, the number of papers and patents published with the keywords “Electrochemical” and “Aptamer” has increased year by year since 2012 (data for 2022 is up to March).

This paper reviews the application of the E-apt sensor for the detection of heavy metal ions in recent years and systematically introduces the working principles of electrochemical sensors with different configurations of various nanomaterials. A Schematic diagram of nanomaterial modified aptamer-based electrochemical sensor are briefly illustrated in Figure 3. The strengths and limitations of this technology are summarized and its future challenges and application trends are envisioned.

## 2. Application of Nanomaterials in Aptamer-Based Electrochemical Sensors

In recent years, the successful synthesis of various nanomaterials has attracted significant attention. Their unique physical and chemical properties, including high surface area/volume ratio, high reactivity, size dependence, and high functionalization, make them widely used in chemical analysis, therapy, diagnosis, and food safety [20,21,22,23,24]. In addition to the binding activity of the aptamer, the reason why the electrochemical aptamer sensors can detect heavy metal ions as low as fM is largely attributed to the rational use of nanomaterials. Sensing platforms are formed by immobilizing aptamers on nanomaterial surfaces through intermolecular forces or catalyzing chemical reactions in sensors as sensitizers to enhance electrical signals and improve the specificity and sensitivity of sensors [25]. Due to their recognized properties, the coupling of aptamers on nanomaterials has great potential to form biosensor platforms.

Gold nanoparticles, carbon nanotubes, graphene, quantum dots, and metal-organic frameworks are common and basic nanomaterials used in E-apt sensors. These materials can be mixed or coupled with other substances to create new composite materials. They have shown many advantages such as high specific surface area, good biocompatibility, high electrical conductivity, high magnetic properties, and unique electro-optic and physicochemical properties [26,27,28]. In addition to being used as sensitizers, most nanomaterials in E-apt sensors are used to fix aptamers to form couplings with a better capture effect. Researchers often use couplings as probes to obtain better detection results. Figure 4 indicates how popular nanomaterials (shown as their microscopic morphology) are coupled with aptamers.

### 2.1. Gold Nanoparticles

Among the aptamer-based sensors developed, metal nanoparticles such as gold nanoparticles (AuNPs) are the most common advanced materials. They have high stability and oxidation resistance and can be prepared through various physical and chemical methods, among which the most classical preparation method was proposed by Turkevich et al. in 1951 [29]. The high surface-volume ratio and excellent optical properties of AuNPs contribute to the high sensitivity and selectivity of the target detection. The optical properties of AuNPs are largely dependent on the size, shape, and aggregation state of nanoparticles. Colloidal AuNPs are usually red or pink and turn purplish-blue when aggregated, forming the basis of colorimetric detection [30]. AuNPs are also conductive and fluorescent and, thus, have the ability to respond to external electrochemical and optical stimulation.

One of the most common approaches to using AuNPs in electrochemical sensors is to anchor aptamers. The aptamer is usually fixed by the force of the Au-S bond; Yuan et al. [31] used this method to develop a sensor for detecting Pb^2+^ and Cd^2+^ at the same time. Due to the specific binding of aptamers to metal ions, methylene blue or ferrocene labeled aptamers were competitively separated from the gold electrode, resulting in a weaker electrochemical signal. The detection limits of Cd^2+^ and Pb^2+^ were 89.31 and 16.44 pM. Deng et al. [32] deposited L-cysteine and AuNPs layers on the electrode surface successively providing a large surface area to anchor many sulfur-capped auxiliary probes through the mercaptan-gold interaction. Additionally, to fix the aptamer, Liu et al. used chitosan to attach AuNPs to the electrode to enhance electron transfer, thus improving the sensitivity of the sensor [33].

In addition to being used alone, AuNPs are also used as composite materials coupled with other metal ions. The composite materials tend to enhance the electrochemical signal so that even subtle changes can be detected. Silver and gold alloy nanoparticles (Ag-Au alloy NPs) promise to create inexpensive and stable electrochemical sensors [34]. This is because silver-gold alloy nanoparticles have a large specific surface area and good electrical conductivity which can act as a conductive center and promote electron transfer, thus trapping more heavy metal ions on the electrode. Zhao et al. [35] used DNA enzyme functionalized (Ag-Au alloy) NPs to form core-shell nanoparticles, which can be used as signal tags. Because the core-shell structure increases the specific surface area and catalyzes the reaction, the electrochemical signal can be enhanced. Ag-Au Alloy NPs were also used to modify GCE and the resulting sensor had higher sensitivity and reproducibility. Miao et al. [36] used Fe_3_O_4_@AuNPs to carry a DNA probe, and the other two were labeled with independent electrochemical substances which can detect both Hg^2+^ and Ag^2+^. Due to their excellent signal amplification, AuNPs have been one of the most used transduction materials to construct E-apt sensors for the analysis of food and water contaminants.

### 2.2. Carbon Nanotubes

Carbon nanotubes (CNTs) are a kind of carbon nanomaterial with unique electrical transmission properties and a large specific surface area, with excellent chemical, mechanical, and thermal stability, and direct binding with other molecules. Therefore, they have received great attention in biosensor design [37,38]. They can be thought of as cylindrical tubes made of one or more sheets of graphene rolled and folded. Nanotubes formed from a single sheet of graphene are called single-walled nanotubes (SWNT), with diameters between 0.4~2 nm, and multi-walled carbon nanotubes(MWCNTs) are 2~10 times larger [39]. Graphene contains SP^2^ hybrid carbon atoms, and nucleic acids can easily be attached to the surface of the nanotubes through π-π bonds, so CNTs are often used to carry ssDNA or RNA. In 2005, So et al. [40] constructed a biosensor using single-walled CNTs by using aptamer as a molecular recognition element for the first time. Since then, the potential of CNTs in sensing platforms has been gradually explored.

It is more common to modify electrodes with MWCNTs and then fix aptamers on the surface of MWCNTs. Zhu et al. [41] used carboxylic acid group functionalized MWCNTs to fix aptamers. Because the material has a large surface area and good charge transferability, the DNA attachment amount and sensor performance could be significantly improved. The sensor could detect Pb^2+^ in the range of 5.0 × 10^−11^~1.0 × 10^−14^ M. Additionally, Zou et al. [42] covalently immobilized aptamers on carboxylic functionalized MWCNTs by EDC/NHS chemistry. CNTs can not only carry aptamers in sensors but also provide binding sites for the reactions between aptamers and other nanomaterials. Rabai et al. [43] modified the electrode surface with CNTs, then deposited AuNPs on the composite electrode. Finally, with the aptamer fixed, the sensor had a detection limit of 0.02 pM for Cd^2+^. He et al. [44] developed an electrochemical sensor with a Zn_3_(PO_4_)_2_ modified aptamer and found that the sensor with MWCNTs had higher sensitivity.

### 2.3. Graphene

Graphene is a two-dimensional carbon nanomaterial with SP^2^ hybrid connected carbon atoms tightly packed into a single layer honeycomb lattice structure. Due to its excellent electrical properties, graphene has attracted great interest in recent years. Graphene has a unique type of structure that also gives it features not found in many other nanomaterials; its large specific surface area makes it an ideal candidate for immobilizing large numbers of functionalized metal oxides and noble metal nanoparticles [45]. Exceptional carrier mobility offers great prospects for nanoscale applications, such as electronic devices and chemical/biological sensors. In addition, its high electron transfer efficiency and wide electrochemical window make graphene an ideal material for constructing a highly sensitive E-apt sensor [46]. In recent years, graphene as a REDOX molecule in electrochemical sensors has attracted great attention.

The most common use of graphene in sensors is to modify electrodes. Zhang et al. [47] developed an Hg^2+^ detector for a graphene-fixed aptamer probe with a detection limit of 5 pM. Using graphene-modified electrodes, the modified electrode surface can be reused. Similar to AuNPs, it is more likely that graphene will be coupled with other nanomaterials to form new composite materials for better modification. Hai et al. [48] modified the electrode with AuNPs coupled with graphene. The aptamer was self-assembled on the electrode, and the detection limit of Pb^2+^ was as low as 3.8 pM. Jiang et al. [49] designed a new nanometer composite material consisting of TiO_2_, AuNPs, and nitrogen-doped graphene. The composite exhibits excellent optical properties, increasing the exciton lifetime and improving the charge transfer photocurrent intensity to 18.2 times higher than that of original TiO_2_. Wang et al. [50] synthesized another new material modified electrode by using CS instead of AuNPs for the detection of Pb^2+^. The modified electrode showed good repeatability, stability, and specificity to other interfering metal ions. Li et al. [51] prepared a sensor for mercury ions based on a perylene-3, 4, 9, 10-tetracarboxylic acid/graphene oxide (PTCA/GO) and quercetin-copper(II) complex. They efficiently promoted the separation of photoexcited carriers and enhanced the photocurrent.

### 2.4. Quantum Dots

Quantum dots (QDs), also known as semiconductor nanocrystals, are mixtures of cadmium and selenium or tellurium with nanoscale clusters with diameters of 1–20 nm [52]. Due to their unique properties, such as high quantum yield, excitation-dependent emission, surface modification versatility, and long-term photostability [53,54], QDs have been used in many research fields, especially in nanoelectronics, optoelectronics, and biological analysis [55]. The physical size of the nanocrystals determines the wavelength of the emitted fluorescence, so multiple analyses can be performed using a single excitation source. With the rapid development of nanotechnology, such nanomaterials have been widely used to improve the sensitivity of sensors. When they are incorporated into the design of biosensors, they can be used as tags, as part of a signal sensor [56].

QDs are mainly used in electrochemical luminescence sensors and photoelectric electrochemistry sensors. When the aptamer reacts with the corresponding target, the brightness of the QDs will also change due to the resonance energy transfer, thus improving the sensitivity of the sensor. CdTe QDs are most frequently used in sensors; Shi et al. [57] developed a new method for the detection of Pb^2+^ based on the sensitization effect of CdTe QDs. When the target is present, the labeled QDs close to the electrode surface produce a sensitization effect and the photocurrent intensity is enhanced. Feng et al. [58] modified CdTe QDs with MIL-53 and determined Hg^2+^ and Pb^2+^ simultaneously by the ECL method with good recovery. Except for CdTe QDs, there are many other types of QDs, such as CdS and nitrogen-doped graphene (NG) [59,60], and their principle of action is roughly the same. When the conformation of the aptamer changes, the distance between the QDs and the electrode alters, and the electrical signal is either enhanced or weakened. This property makes QDs widely used to design E-apt sensors for food and water analysis.

### 2.5. Metal-Organic Frameworks

In recent years, more and more metal-organic frameworks (MOFs) have been found to effectively coordinate polymer nanomaterials and have received extensive attention as effective quenching materials. MOFs are a kind of crystalline nanomaterial composed of metal ions and organic ligands. Due to their water dispersibility, adjustability, biocompatibility, low cost, controllable shape, ultra-high porosity, and high specific surface area [61], MOFs have been increasingly used in biosensors, electrocatalysis, energy storage, and conversion. The large specific surface area and ultra-high porosity of MOFs provide more reaction sites for aptamers and targets. Organic ligands with rich functional groups make MOFs easy to be functionalized with various molecules and materials [62]. In addition, MOF compositions take a variety of forms (e.g., nanosheets, cages, tubes, rods, cubes, etc.) and can be easily adjusted according to the selection of various organic connectives and metal ions [63].

Aptamers are securely fixed in the MOFs by encapsulation. The main framework of MOFs can facilitate various interactions with analytes through functional groups in organic ligands, thus achieving high sensitivity and high selectivity recognition. Therefore, MOFs can be used as signal probes for different detection methods. Zhang et al. [64] used a Zr-based MOF embedded with three kinds of aptamer. Ling et al. [65] prepared streptavidin functionalized zirconium porphyrin MOF (PCN-222@SA) using a covalent method as a signal nanoprobe. Introducing this signal nanoprobe into the sensor surface significantly amplified the electrocatalytic current. Zhang et al. [66] synthesized a core-shell nanostructured composite material composed of Fe (III)-based MOF (Fe-MOF) and mesoporous Fe_3_O_4_@C nanocapsules (Fe-MOF@mFe_3_O_4_ @MC) that exhibited excellent electrochemical activity, water stability, and high specific surface area, resulting in strong biological binding to heavy metal ion targeting aptamer chains.

## 3. Electrochemical Techniques

E-apt sensors have proven to be effective tools for detecting heavy metal ions. They are cheap, portable, and easy to operate, making them popular with researchers. The concentration of the target is detected by collecting the changes in voltage, current, conductivity, impedance, and other electrochemical signals [67,68]. Table 1 summarizes the application of the E-apt sensor in the detection of heavy metal ions in recent years. According to different electrochemical principles, electrochemical characterization and detection methods can be divided into cyclic electrochemical impedance spectroscopy (EIS), differential pulse voltammetry (DPV), square wave voltammetry (SWV), photoelectric electrochemistry (PEC), electrochemical luminescence (ECL), and other methods [69]. When the analyte is around, the target-induced signal is generated and recorded as the corresponding electrochemical signal, which can be detected in a few minutes with relatively high sensitivity. Sensors with different principles use different nanomaterials and sensitizers, as well as different construction methods. Various electrode structures, construction methods, and responses to inspection objects are summarized in Figure 5.

### 3.1. Electrochemical Impedance Spectroscopy

EIS is an effective technique for detecting complex formations on the electrode surface by detecting interface phenomena [70,71].

It measures the resistive and capacitive properties of the electrode upon perturbation with a small amplitude AC excitation. The frequency is varied over a wide range to generate the impedance spectrum, and the steady-state electrical impedance of the electrode/electrolyte interface is measured over an appropriate frequency range by applying a small sinusoidal voltage [72]. Impedance can be understood as the ratio of the voltage phasor to the current phasor of the system. When the impedance changes, it indicates that the detection target has been combined with the biometric element fixed on the electrode surface. EIS is usually able to distinguish between two or more electrochemical reactions occurring at the same time, identify diffusion-limiting reactions, mathematically evaluate experimental results using equivalent circuits (EEC), and reliably provide quantitative electrochemical data.

AuNPs are mainly used as electrodes and modification materials in EIS sensors [73,74,75,76]. When gold is used as the electrode, the aptamer is usually fixed on the surface of the gold electrode to generate a huge charge transfer resistance (Rct). When the corresponding target substance is added, the aptamer falls off from the surface of the gold electrode and combines with the target to produce an electrical signal transformation. Gu et al. developed an ultra-sensitive As^3+^ biosensor based on the hybridization chain reaction and RecJf exonuclease catalyzed reaction. In the presence of As^3+^, the aptamer specifically binds to As^3+^, resulting in DNA dissociation. The release of the hybrid chain reaction (HCR) product significantly reduced the Rct, and the detection limit was 0.26 nM. Moreover, the sensor has good selectivity. Even though the concentration of potential interfering ions is ten times higher than that of As^3+^, the change in Rct is negligible and only sensitive to As^3+^ [77]. Rabai et al. modified the surface of the gold electrode by a diazonium salt (CMA) electrochemical reduction method for the fixation of the aptamer. When Cd^2+^ was present, the aptamer changed from a random coil structure to a complex. This interaction blocks electron transfer, increasing the surface resistance, which is proportional to the concentration of Cd^2+^ in the sample [78]. In addition to traditional gold electrodes, the use of inkjet-printed gold electrodes as a reliable method for the detection of trace Hg^2+^ in water and organic solvents was first proposed in 2019. They applied water droplets of gold ink to a substrate and sintered it under the right conditions to produce printed gold electrodes. The aptamer was fixed to the electrode by the force of disulfide bonds, and then the interface was assembled layer by layer using impedance spectroscopy (PEIS) via RCT under optimal manufacturing conditions. With the increase in layers, RCT increases gradually. The interfacial resistance increases from an average of 20.6 U to 144.5 U when the aptamer is fixed on the surface. With the addition of Hg^2+^, the ssDNA aptamer will change its secondary conformation by folding into a hairpin structure, establishing a bridge between the two thymidine residues, and forming a base pair, with a significantly reduced RCT. Figure 6a shows evidence of a directly proportional relationship between the response variable (RCT) and the target concentrations under optimal conditions. The working principle, signal strength, and linearity are also shown in Figure 6. The sensor has good stability and significant repeatability under harsh conditions and can detect Hg^2+^ at 0.005 ppM in organic solvents. Even when compared with high concentrations of cadmium, lead, and arsenic (50 ppM), it still has good selectivity [79]. 

AuNPs are commonly used to modify various carbon-based materials such as graphene oxide (GO), CNTs, etc. The high porosity of GO provides a large number of reaction sites for the bonding of AuNPs with aptamers. Wang et al. [80] used gold to modify porous GO (Au@p-rGO) and used it to fix substrate aptamers. Gold-modified GO (AuNPs@GO) was attached to the complementary chain as a signal probe; the current signal was obtained by recording the electrocatalytic conditions of H_2_O_2_. With the increase in Pb^2+^ concentration, the current response showed signal attenuation, and the lower limit of detection was 1.67 pmol/L. Rabai et al. [43] first dispersed CNTs in chitosan (CS) to modify a glassy carbon electrode (GCE). Chitosan improves its solubility and biocompatibility, and AuNPs are then electrically grafted onto the CNT-Cs/GCE. The synergistic effect of the two will make electron transfer easier, have good electrical conductivity, high surface electrical activity, and a detection limit as low as 0.02 ppm. Because the aptamer does not react with other heavy metal ions and the two-dimensional structure does not change, the sensor also has good selectivity. Yadav et al. [81] used silver (Ag)-Gold (Au) alloy nanoparticles (NP)-aptamer on a modified glassy carbon electrode (GCE) to detect Pb^2+^. Ag-Au bimetallic nanoparticles have electronic polarity which attracts the charged adsorption and promotes the attachment of adsorption to the loading platform on the surface of GCE. A large number of binding sites can improve the sensitivity and stability of the electrode and reduce its detection limit.

### 3.2. Differential Pulse Voltammetry

The working principle of DPV is the constant voltage pulse amplitude overlayed on the step potential; immediately before each potential change measuring the electric current. The implementation response is the pulse strength at the beginning of the two currents between the resulting peak response; the difference between the two currents minimizes the amount of capacitance current, generating a higher signal-to-noise ratio of voltammograms [82]. DPV is a universal technique that can be used for both quantitative chemical analysis and the study of the mechanism, kinetics, and thermodynamics of chemical reactions. DPV is very sensitive and can routinely detect analytes at a part per billion level with a resolution higher than that achieved by cyclic voltammetry (CV), therefore, DPV is superior to CV when higher selectivity is required. 

Reduced graphene (rGO) is commonly used in DPV sensors. It can be coupled with other nanomaterials as signal amplifiers or directly used as electrodes, which can significantly change the electrochemical properties. Luo et al. [83] used Fe_3_O_4_/rGO nanocomposite as a signal amplifier and many directional platinum nanotube arrays (PtNAs) crystallized in situ on flexible electrodes as sensing interfaces; a Hg^2+^ sensor was prepared. The schematic illustration of the assembly process and the detection strategy is shown in Figure 7. Due to its large surface area, it facilitates electrochemical performance and fixation of captured DNA (cDNA) and reporter DNA (rDNA). In the presence of Hg^2+^, part of the junction DNA binds closely to cDNAs through a thymine nucleotide pair (T-Hg^2+^-T). The Fe_3_O_4_/rGO nanoprobes attached to rDNAs were then fixed to the electrode by matching the remaining linker DNA with the rDNAs. Under optimal conditions, the Hg^2+^ aptamer sensor showed synergistic amplification performance with a linear range from 0.1 nM to 100 nM and a detection lower limit of 30 pM. When the heavy metal ions were not mercury ions, the aptamers could not undergo conformational changes through thymine nucleotides so this sensor had good selectivity. In addition, the E-apt sensor also showed reliable performance in the detection of real lake water samples.

When used as an electrode, aptamers can be fixed on rGO electrodes (ERGO) by π-π interaction. The ERGO electrode modified by aptamers can improve the value of Rct. Lee et al. [84] fixed the aptamer probe (Apt) labeled with methylene blue (MB) and part of its complementary DNA (cDNA) on the ERGO electrode to form the aptamer double-stranded structure, which blocked the effective electron transfer of MB to the electrode. After adding Cd^2+^, the aptamer unlocked the link and released the cDNA. This can quantitatively promote the electron transfer efficiency of MB, leading to the enhancement of electrochemical signals. Su et al. [85] also used this strategy to detect ultra-trace Pb^2+^, as the existence of Pb^2+^ could make Apt fold into a G-quadruplex structure. The formation of the G-quadruplex leads to the separation of Apt from ERGO/GCE, which changes the REDOX current of MB labels with a detection limit of 0.51 fM. The sensor was tested in the presence of various metal ions (Cd^2+^, Co^2+^, Ag^+^, Cu^2+^, Mg^2+^, Ni^2+^, Zn^2+^, and Fe^2+^), but only Pb^2+^ resulted in a significant change in voltammetry response and had good repeatability. In addition to MB, toluidine blue (TB) molecules are also commonly used for electron migration in sensors, and the peak current of TB interacting with double-stranded DNA (dsDNA) is higher than that of single-stranded DNA (ssDNA). Ding et al. [86] used the composites of Au nanoparticles and a Polypyrene (Au@Py) modified screen printing electrode to amplify the current signal, fix the complementary chain on the electrode, and combined it with the aptamer to form a double chain structure. When Pb^2+^ was combined with an aptamer, the double chain structure was destroyed, and the peak current decreased continuously. Ma et al. [87] developed a sensor for Hg^2+^ ultra-sensitive determination, also using TB to characterize electron migration, using mesoporous silica nanocontainers (MSNs) as containers. MSNs have a rich porous structure that can trap TB molecules using AuNPs to link specific ssDNA. Hg^2+^ induces ssDNA to form a hairpin structure and the stored tuberculous molecules are released from MSNs. The electron transfer signal of TB was stably detected by micro DPV, which was correlated with the concentration of Hg^2+^, with a low detection limit of 2.9 pM. Jin et al. [88] developed an electrochemical adaptive sensor for Pb^2+^ detection using porous carbon (PCs) loaded platinum nanoparticles (PtNPs) to catalyze the hydroquinone-H_2_O_2_ system in the form of simulated enzymes. PtNPs@PCs were fixed on the electrode surface by the specific binding of streptavidin and biotin and catalyzed the oxidation of hydroquinone in the presence of Pb^2+^ and H_2_O_2_. The resulting electrochemical signal was dependent on the concentration of Pb^2+^.

### 3.3. Square Wave Voltammetry

SWV is a pulsed method in which the waveform is defined by step potential, amplitude, and period. The excitation signal used consists of a symmetrical square wave pulse superimposed on the stepped waveform, where the forward pulse of the waveform corresponds to the step to form a rectangular wave [89]. The current intensity is obtained at the end of each applied pulse during the potential sweep cycle. The difference between positive and negative pulse currents is recorded at the pulse time, and the difference between the forward and reverse currents over the same period is called the net current. The advantages of SWV are improved speed, background discrimination and sensitivity, and good discrimination of non-Faraday or charging currents.

In recent years, screen-printed electrodes (SPE) have been widely used in the design of biosensors. Their diverse functions, low cost, and easy use have attracted great interest. Moreover, the SPE can be modified with nanomaterials to enhance its electrochemical performance. Fakude et al. [90] first proposed an E-apt sensor for Cd^2+^ detection based on flexible polyester SPE. A Cd^2+^ join makes the fit body configuration change; an iron/ferrocyanide REDOX probe can then go more easily through the electrode surface, and the carbon nanofibers (CNF) can promote a simple electron transfer reaction. CNF after acid treatment on the polyester SPE is increased by the electrical activity of the electrode surface area and catalytic REDOX process of the iron/ferrocyanide, which enhances electron flow. Fakude [91] also modified the SPE with carbon black nanoparticles and AuNPs. The strategy is to deposit AuNPs by CV after modifying the electrode with carbon nanoparticles, so the Faraday current can increase up to 80% with a detection limit of 0.14 ppb. This strategy has also been used to detect As^3+^ in water [92], but the electrode used in this sensor is a glassy carbon electrode (GCE). The interface properties of the electrode are characterized by charge transfer resistance and double layer capacitance. The presence of nanoparticles on the detection limit GCE shows a significant decrease in the Rct value. The detection limit was 0.092 ppb and the specificity was well.

Si et al. [93] proposed an electrochemical biosensor based on aptamer-terminal deoxynucleotidyl transferase (TdT), which catalyzed the continuous polymerization of adenine bases, resulting in the formation of long polyA which enabled Si-DNA to be anchored on the electrode surface and enhanced electrical signals. The introduction of Hg^2+^ leads to the formation of the T-Hg^2+^-T complex, which prevents TdT from forming polyA, resulting in the absence of Si-DNA on the electrode surface and the decrease in the electrochemical signal.

### 3.4. Photoelectric Electrochemistry

PEC is a kind of photoelectric analysis technology developed gradually based on electrochemistry in recent years. It is a highly sensitive and fast analysis method. PEC converts chemical energy into electrical energy by using light as the excitation source, and the photocurrent generated is used as the detection signal. It has the advantages of low background noise and high sensitivity and has been widely used in chemical synthesis, catalysis, and biological analysis [94].

In the process of PEC sensor construction, photoactive materials play an indispensable role, and their properties directly determine the performance of the sensor [95]. Various materials such as TiO_2_, ZnO, CdSe, and CdS are used to manufacture PEC sensors. ZnO nanomaterials are a kind of N-type semiconductor and have attracted extensive attention due to their advantages of low cost, good chemical stability, and excellent electrical and optical properties. Cao et al. [55] modified ZnO nanosheets on an ITO electrode, and then modified CdS nanoparticles on the surface of ZnO nanosheets to form a CdS/ZnO-sensitized structure through continuous ion layer adsorption and the reaction of Cd^2+^ and S^2−^. Then, CdSe QDs were introduced into the sensing system through a hybridization reaction, forming a double co-sensitization structure, realizing high selectivity, high sensitivity, and high stability detection of Pb^2+^. Niu et al. [96] designed a ZnO and Reduced graphene oxide (ZnO-RGO) nanocomposite as a photoactive material, adding AuNPs to further enhance the electrical conductivity. Moreover, AuNPs can anchor the aptamer and its complementary chain to form a double chain structure in which MB can amplify the current response. When the sensor captures Cd^2+^, the aptamer and its complementary chain break, and MB is separated from the electrode surface, reducing the photocurrent response, resulting in a detection limit of 1.8 × 10^−12^ mol/L. Niu also designed ZnO-TiO_2_ nanocomposites as photoactive substrates and covered them with gold nanochains. One part of the aptamer was connected to the gold nanochain, and the other part was coupled to graphite-like Carbon Nitride (G-C_3_N_4_). When Cd^2+^ was detected by the aptamer sensor, the aptamer formed a stable hairpin structure, and the signal sensitizer G-C_3_N_4_ was closer to the electrode, making the changes in the photocurrent signal more sensitive. The detection limit was 1.1 × 10^−11^ mol/L, slightly lower than that of the first method. Both sensors have significant specificity because the aptamer usually reacts only after contact with its corresponding target, resulting in changes in secondary structure. Niu [97] developed a PEC sensor for Pb^2+^, again using AuNPs as a fixed aptamer, and CdS-TiO_2_ as a photoactive material, The difference is quercetin-copper(II) complex as intercalator and electron donor. The detection limit was 1.6 × 10^−12^ mol/L, which was satisfactory.

### 3.5. Electrochemiluminescence

Electrochemiluminescence (ECL) has received considerable interest in the development of an ultrasensitive detection technique in recent years. It works by bringing the system or component of the electric biomass into an excited state through electron transfer and then returning it from the excited state to the ground state to produce a chemiluminescence phenomenon. It combines the advantages of both electrochemical and chemiluminescent biosensors, with relatively low cost, simplicity, rapidity, and high selectivity [98,99].

Ruthenium (II) tris (bipyridine) (Ru(bpy)_3_^2+^) and its derivatives remain the most popular ECL reagents due to their recyclability, high quantum yields, and suitability in different pH levels, It interacts with the electrode surface to produce a strong ECL signal, usually by using hairpin DNA to interact with Ru(bpy)_3_^2+^ away from the electrode surface, thereby reducing the ECL signal [100]. According to this principle, Strand Displacement Amplification (SDA) is also widely used in the construction of ECL sensors. SDA was proposed and improved in 1992, relying on the combination of the strand-displacing polymerase and the nicking endonuclease to generate an exponential accumulation of single-stranded DNA (ssDNA) [101]. Zhu et al. [102] prepared a sensor for As^3+^ based on SDA technology. By using polydopamine nanospheres (PDANS) as inhibitors, hairpin DNA was constrained by PDANS and the SDA process was inhibited. Ru(bpy)_3_^2+^ as an ECL probe could diffuse the ITO electrode surface and generate a strong ECL response. However, the presence of As^3+^ makes hairpin DNA no longer constrained and triggers the SDA process with the help of polymerase and incisor endonuclease to generate dsDNA, which interacts with Ru(bpy)_3_^2+^ to form the dsDNA- Ru(bpy)_3_^2+^ complex. Due to electrostatic repulsion, it is difficult for the complex to approach the ITO electrode surface, resulting in a low ECL response, the detection limit is 1.2 × 10^−3^ ppb. This strategy has also been applied to the detection of Cd^2+^ [103]. The difference is that Xu et al. used magnetic Fe_3_O_4_-GO nanosheets to constrain hairpin DNA, and the detection limit was 1.1 × 10^−4^ ppb. Ma et al. [104] did not use an SDA reaction for DNA amplification but hybridized a ruthenium complex with an aptamer and its complementary chain. The combination of Hg^2+^ and T-T mismatch induced adaptive folding and compression of the aptamer, keeping the Ruthenium complex away from the electrode and weakening the ECL signal. Li et al. [59] functionalized Ru(bpy)_3_^2+^ with 3-aminopropyltriethoxysilane and mixed it with silica nanoparticles and graphene quantum dots to form an ECL composite material for mercury ion detection, which has good detection performance.

In addition to Ru(bpy)_3_^2+^, the ECL resonance energy transfer (RET) of QDs and precious metal nanoparticles (such as AuNPs) is also considered to be a sensitive and reliable analysis technique, but their ECL intensity is usually lower than Ru(bpy)_3_^2+^. Wang et al. [105] synthesized cadmium sulfide QDs doped with lanthanum ions and designed an ECL sensor based on the QDs and AuNPs. The surface plasmon resonance of AuNPs enhanced the strength of ECL. Secondly, in the presence of Hg^2+^, oligo-base pairs change from linear chains to hairpins. The realized ECL quenched, and finally, after incubation with TB, produced a strong and stable transfer, which resulted in the eventual recovery of the ECL signal. The detailed process is shown in Figure 8.

## 4. Prospects and Challenges

This review provides an overview of the recently developed E-apt sensors and the use of different nanomaterials in sensors. These sensors can be applied to the analysis of various food and water contaminants. By using aptamers with high affinity and specificity for the target and one or more effective signal amplification steps, most sensors exhibit good sensor behavior, including high sensitivity and selectivity.

However, the practical application of electrochemical sensors is still in the preliminary stage, and most heavy metal detection is carried out under laboratory conditions. Results obtained in the laboratory are difficult to verify with results from real samples; there is still a long way to make developments in the practical identification of heavy metal ions. Therefore, further research perspectives that must be taken into account in this area are as follows. The efficiency of the sensor largely depends on the affinity of aptamers, so screening new aptamers with high affinity is the first prerequisite to constructing excellent sensors. In addition, adaptation is also highly dependent on nanomaterials. The design of highly active nanomaterials with long-term stability and reproducibility is the main goal of future efforts. Future research can be carried out towards the advancement of E-apt sensors based on advanced technologies for the multi-residue determination of heavy metal ions in various environments.

Electrochemical sensors have bright application prospects, but there are still some problems that need to be solved Firstly, the binding efficiency of the aptamer and heavy metal ions should be considered. The detection ability of the aptamer sensor depends largely on whether the aptamer can detect the existence of the target sensitively. If the aptamer and the target are combined inefficiently, it is difficult to achieve the ideal detection effect. Secondly, the efficiency of the SELEX method and the specificity of the selected aptamer still need to be improved. Because the aptamer selection method is time-consuming, low cost, and with low efficiency, there are not many aptamers on the market for the detection of heavy metal ions which greatly hinders the development of aptamer-based sensors. Third, because E-apt sensors usually need to use nanomaterials to improve their sensitivity, the cost of nanomaterials is affected by the price of nanomaterials. The development of low-cost nanomaterials with significant effects is also crucial to the construction of sensors. Finally, the survey found that most of the aptamer-based sensors are often only for single heavy metal ion detection and usually, more than in the sample, there are multiple kinds of heavy metal ion pollution. The development of multiple target detection sensors then has important practical significance and broad application prospects. 

In conclusion, an aptamer-based biosensor design provides a promising method for the fast and on-site monitoring of heavy metal ions in food safety, but further research and development are still needed. 

## Figures and Tables

**Figure 1 foods-11-01404-f001:**
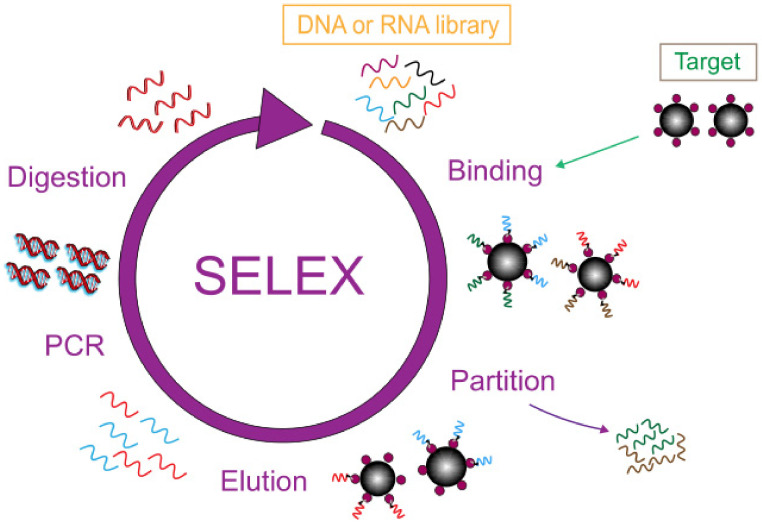
SELEX flowchart.

**Figure 2 foods-11-01404-f002:**
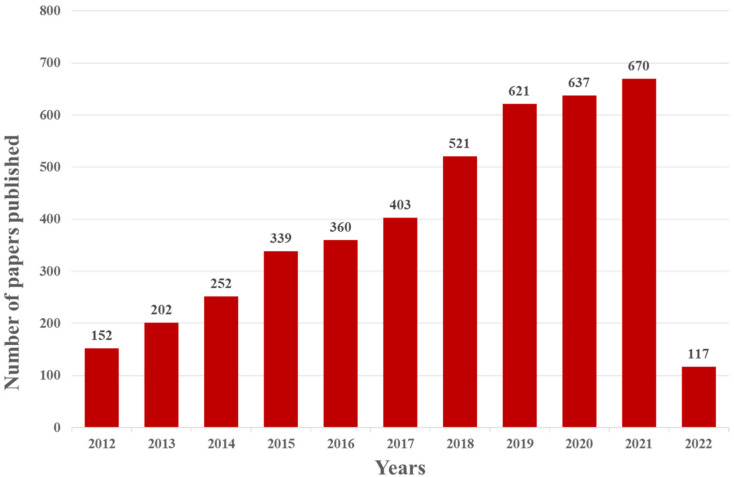
Web of Science report for the number of indexed papers and patents about the application of the E-apt sensor (keyword: “Electrochemical” and “Aptamer” accessed on 27 March 2022).

**Figure 3 foods-11-01404-f003:**
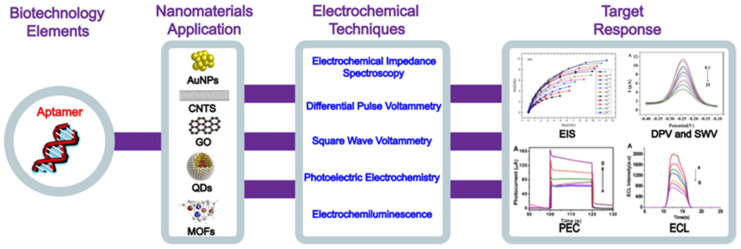
Schematic diagram of nanomaterial modified aptamer-based electrochemical sensor.

**Figure 4 foods-11-01404-f004:**
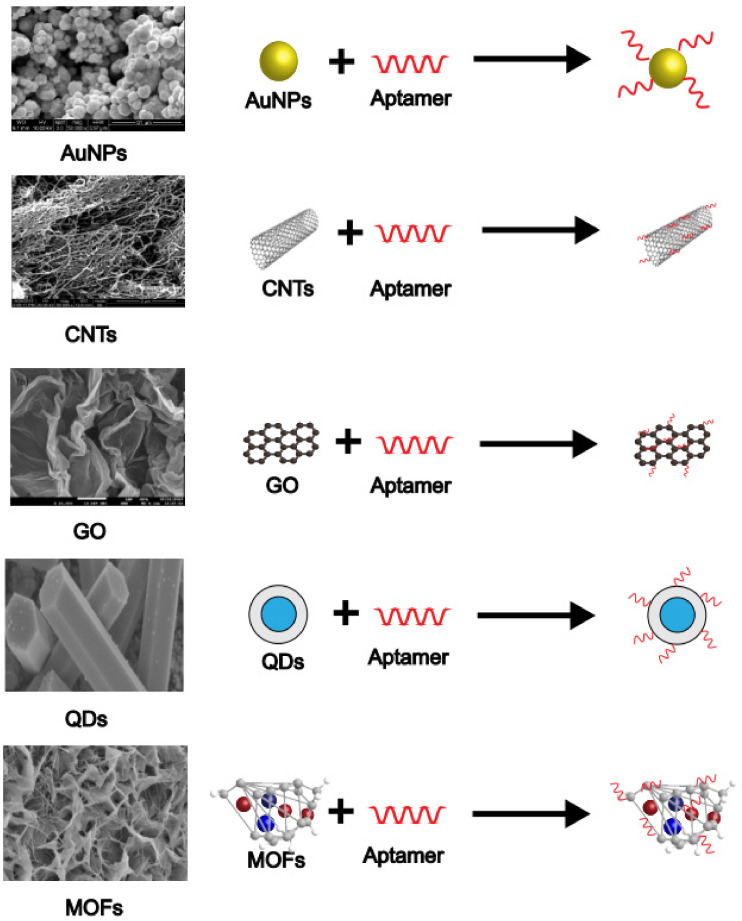
SEM images and combination modes of various nanomaterials.

**Figure 5 foods-11-01404-f005:**
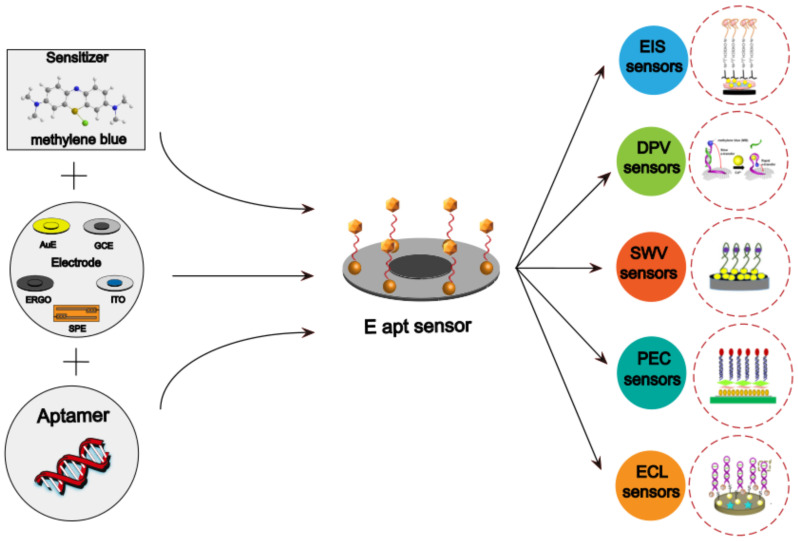
Electrochemical sensor technology is based on aptamers and different electrodes.

**Figure 6 foods-11-01404-f006:**
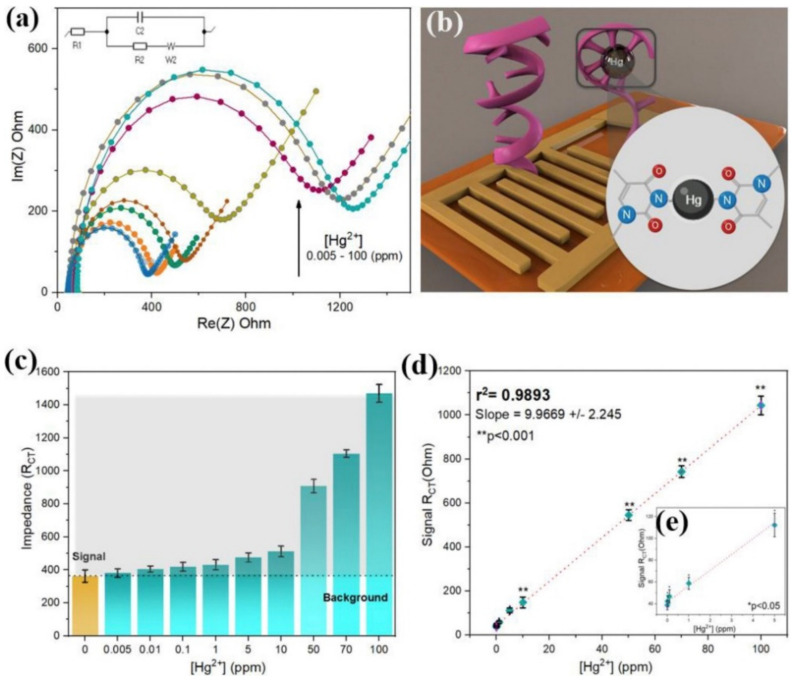
Inkjet−printed electrochemical apt sensor performance analysis; (**a**) Analytical Faradaic impedance response under optimal fabrication conditions; (**b**) Working electrode diagram; (**c**) The bar chart represents the overall reaction, and the shaded area represents the signal strength after subtracting the background; (**d**) Linear correlation between signal and target concentration; and (**e**) All the results correspond to the mean value from 5 independent replicates and the error bars represent 1 SD from the mean. Reprinted with permission from ref. [79]. Copyright 2019 Elsevier.

**Figure 7 foods-11-01404-f007:**
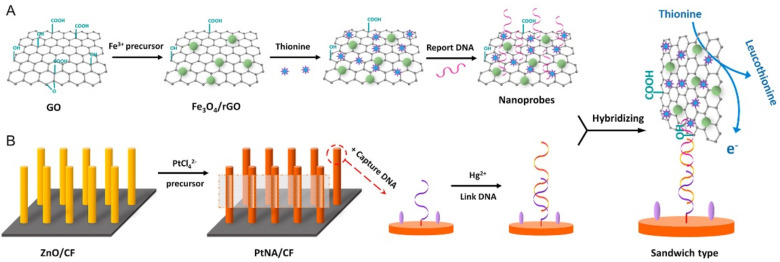
Schematic illustration of the assembly process and the detection strategy; (**A**) Preparation of nano-probe; (**B**) Modification of flexible electrode. Reprinted with permission from ref. [83]. Copyright 2018 Elsevier.

**Figure 8 foods-11-01404-f008:**
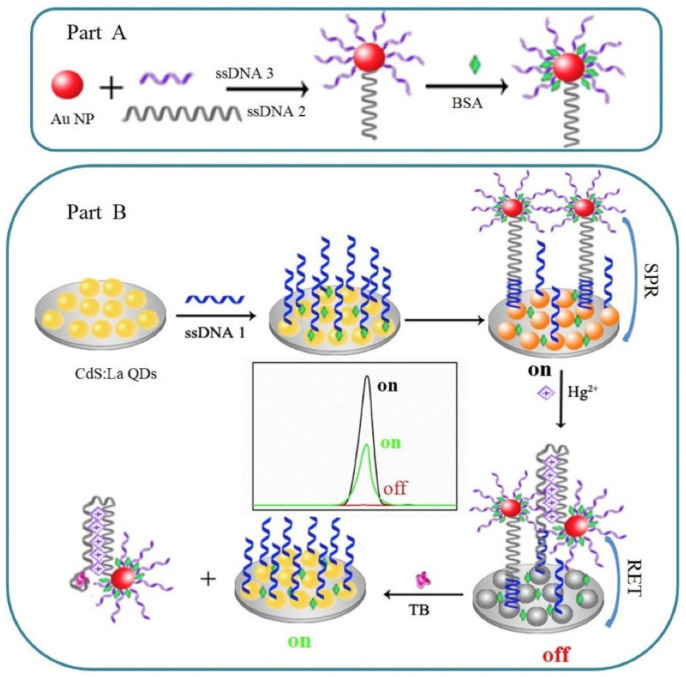
Fabrication of the ECL-RET aptasensor for Hg^2+^ and TB, respectively based on the resonance energy transfer between CdS:La QDs film and AuNPs; (**A**) Preparation of aptamer probe; (**B**) Electrode surface modification and sensor reaction principle. Reprinted with permission from ref. [105]. Copyright 2019 Elsevier.

**Table 1 foods-11-01404-t001:** Application of the electrochemical sensor in the detection of heavy metal ions.

Method	Target	LOD (nM)	Linear Range (nM)	Aptamer Sequence	Sample	Reference
EIS	Hg^2+^	0.071	0.1~50	5′-CCCCCCCCCCCCTTCTTTCTTCCCCT TGTTTGTT-3′	Tap water	[44]
	Hg^2+^	25	25~500	5′-TTTCTTCTTTCTTCCCCCCTTGTTTGT TT-3′	Water	[79]
	Cd^2+^	0.275	1~1 × 10^6^	5’-ACCGACCGTGCTGGACTCTGGACTG TTGTGGTATTATTTTTGGTTGTGCAGTA TGAGCGAGCGTTGCG-3′	River water	[78]
	Pb^2+^	1.67 × 10^−3^	5 × 10^−3^~1	5′-HS-TTTTTTCGATAACTCACTATrAGG AAGAGATG-3′	Serum Water	[80]
	As^3+^	0.26	1.3~6.5	5′-TGATGTTTGTTTACGCATGTGTGAG GAGGCTGGGGTGATGAATCCCAATCCC-3′	Tap water	[77]
	Pb^2+^ As^3+^	2.27 × 10^−3^ 6.73 × 10^−3^	0.01~10.0	5′-CAACGGTGGGTGTGGTTGG-3′ 5′-GGTAATACGACTCACTATAGGGAG ATACCA GCTTATTCAATTTTACAGAA CAACCAACGTCGCTCCGGGTACTTCTT CATCGAGATAGTAAGTGCAATCT-3′	River water	[66]
DPV	Hg^2+^	0.03	0.01~100	5′-SH-(CH_2_)_6_-AAAAATTTCCTTTGCTTT-3′	Lake water	[35]
	Hg^2+^	5 × 10^−3^	0.025~1 × 10^−6^	5′-(NH_2_C_6_)-CTT GCT TTC TGT-3′	Lake water	[47]
	Hg^2+^	0.03	0.1~100	5’-SH-(CH_2_)_6_-ACCGTGTTTGCCTTTGAC CTC-3’	Lake water	[83]
	Hg^2+^	2.9 × 10^−3^	0.01~1 × 10^5^	5′-COOH-CTTCTTCCCCCCCCTTCTTC -SH-3′	River water	[87]
	Hg^2+^	5 × 10^−3^	0.01~500	5′-SH-(CH_2_)_6_-TCATGTTTGTTTGTGGCC CCCCTTCTTTCTTA-Fc-3′	Tap water	[106]
	Hg^2+^	0.33	1~200	5′-Bio-TCTTTCTTCCCTTGTTTGT-3′	Tap water	[107]
	Cd^2+^	5 × 10^−5^	1 × 10^−3^~100	5′-ACCGACCGTGCTGGACTCTGACTG TTGTGGTATTATTTTTGGTTGTGCAGT ATGAGCGAGCGTTGCG-3′	Tap water	[33]
	Cd^2+^	6.5 × 10^−7^	1 × 10^−6^~1	5′-GGGGGGGGACTGTTGTGGTATTATT TTTGGTTGTGCAGT-MB-3′	Valley water	[84]
	Pb^2+^	4.3×10^−9^	1.0×10^−8^~5.0×10^−5^	5′-GGGTGGGTGGGTGGGT-3′	Springwater	[41]
	Pb^2+^	1.6 × 10^−3^	4.8 × 10^−3^~4.8	5′-GGTTGGGCGGGATGGGTG-3′	Tea and Rice	[50]
	Pb^2+^	5.1 × 10^−7^	1 × 10^−6^~1	5′-MB-GGTGGTGGTGGTTGTGGTGGTG GTGG-3′	Tap water	[85]
	Pb^2+^	2.88	2.4~120	5′-GGGTGGGTGGGTGGGT-3′	Soil	[86]
	Pb^2+^	0.018	0.05~1 × 10^3^	5′-GGGTGGGTGGGTGGGTAT-3′	Tap water	[88]
	Pb^2+^	0.312	0.5~50	5′-GGGTGGGTGGGTGGGT-3′	Serum	[108]
	As^3+^	4 × 10^−5^	0.13~130	5′-HS-GGTAATACGACTCATAAGGGA GATGCTTATTCAATTTTACAGAACACCAAGTCGCTTACTTCTTCATCGAGATAGTAAGTGCAATCT-3′	River water	[34]
SWV	Hg^2+^	1.79	10~100	5′-MB-CGCTTTAGATG-3′	Juice	[36]
	Hg^2+^	1 × 10^−4^	2 × 10^−3^~20	5′-SH-AATTCTCTCTTCGACGTTGTGT GTT-3′	Tap water	[93]
	Hg^2+^	0.094	1~5 × 10^3^	5’-SH-(CH_2_)_6_-CTGTTTTCTTTCGGACGA CCCCCCTCGTCCGTTTGTTTTCAG-MB^+^-3′	River water	[109]
	Cd^2+^ Pb^2+^	0.089 0.016	0.1~1000	5′-CTCAGGACGACGGGTTCACAGTC CGTTGTC-Fc-3′ 5′-GGT TGG TGT GGT TGG-MB-3′	Lettuce Orange	[31]
	Cd^2+^	0.014	0.1~5	5′-HS(CH_2_)_6_GGACTGTTGTGGTATTAT TTTTGGTTGTGCAGTATG-3′	Tap water	[91]
	As^3+^	0.7	3.83~766	5′-HS-GGTAATACGACTCATTAGGGAG ATCAGCTTATTCAATTTTACAGAACA ACCAACGTCGCTCCGGTACTTCTTCAT CGAGATAGTAAGTGCAATCT-3′	None	[92]
PEC	Hg^2+^	3.33 × 10^−6^	1 × 10^−5^−1 × 10^−3^	5′-NH_2_-(CH_2_)_6_-TTTTTTTTTTTTTTTTTTTT -3′	Tap water	[51]
	Cd^2+^	1.8 × 10^−3^	5 × 10^−3^~29	5′-GGACTGTTGTGGTATTATTTTTGGT TGTGCAGTATG-3′	Lake water	[96]
	Cd^2+^	0.011	0.03~40	5′-SH-GGACTGTTGTGGTATTATTTTTG GTTGTGCAGTATG-NH_2_-3′	Lake water	[110]
	Pb^2+^	3 × 10^−4^	1 × 10^−3^~5	5′-TTGGGTGGGTGGGTGGGT-3′	Tap water	[49]
	Pb^2+^	1.67 × 10^−5^	5 × 10^−5^~1 × 10^3^	5′-NH_2_-(CH_2_)_6_-TTGGGTGGGTGGGTGGG T-P-3′	Reservoir water	[57]
	Pb^2+^	0.05	0.1~50	5′-NH_2_-(CH_2_)_6_-TTGGGTGGGTGGGTGG GT-3′	Tap water	[60]
	Pb^2+^	1.6 × 10^−3^	5 × 10^−3^~10	5′-SH-GGGTGGGTGGGTGGGT-3′	Soil	[97]
	Pb^2+^	0.34	1~1 × 10^4^	5′-NH_2_-(CH_2_)_6_-TTGGGTGGGTGGGTGGG T-3′	River water	[55]
ECL	Hg^2+^	0.01	0.05~1 × 10^3^	5′-NH2-TTGTTTGTCCCCTCTTTCTTA-(CH2)3-SH-3′	Tap water	[59]
	Hg^2+^	4 × 10^−5^	1 × 10^−4^~0.01	5′-amino-(CH2)6-O-TCTCCAGCGTCGTT TGTTTGCGGGAGCTTTCTTAAATCTCG AGCTAAA-3′	Water	[104]
	Hg^2+^	3 × 10^−4^	1 × 10^−3^~1 × 10^4^	5′-GGTTGGTGTGGTTGGTTCTTTCTT CCCTTGTTTGTT(CH2)6-SH-3′	None	[105]
	Hg^2+^ Pb^2+^	4.1 × 10^−6^ 2.4 × 10^−5^	1.0 × 10^−5^~0.01 1.0 × 10^−4^~10	5′-TTTTTTAAAATTTTTT-SH-3′ 5′-COOH-(CH_2_)_10_-AAAAAAAAAGGGG-SH-3′	Shrimp	[58]
	Cd^2+^	9.7 × 10^−4^	0.26~2.6 × 10^6^	5′-ACCGACCGTGCTGGACTCTGGAC TGTTGTGGTATTATTTTTGGTTGTGCA GTATGAGCGAGCGTTGCG-3′	Extracting solution of sophora	[103]
	Pb^2+^	4 × 10^−8^	1.0 × 10^−7^~0.1	5′-GGTTGGTGTGGTTGG-3′	Soil	[32]
	Pb^2+^	3.82 × 10^−6^	1.0 × 10^−5^~1.0 × 10^−2^	5′-SH- (CH_2_)_6_-TTTTTACCCAGGGTGGG TGGG-TGGGT-(CH_2_)_6_-NH_2_-3′	River water	[48]
	As^3+^	9.2 × 10^−3^	15.3~1.53 × 10^4^	5′-GGTAATACGACTCACTATAGGGAG ATACCAGCTTATTCAATTTTACAGAA CAACCAACGTCGCTCCGGGTACTTCT TCATCGAGATAGTAAGTGCAATCT-3′	Extracting solution of sophora	[102]

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
