# Peer review of "Application of Nanomaterial Modified Aptamer-Based Electrochemical Sensor in Detection of Heavy Metal Ions"

_foods, 2022, doi:10.3390/foods11101404_

Round 1

Reviewer 1 Report

Comments and suggestions for authors

In this paper, the research progress on the application of nanomaterial modified aptamer-based electrochemical sensors for the detection of heavy metal ions in environmental water matrices is reviewed. The application, advantages and challenges of E-apt sensor in heavy metal ions detection are summarized. Overall, the paper is scholarly. However, a few issues should be well addressed.

[1] The review manuscript is well structured with logical flow, however, it has so many grammatical errors. I therefore suggest that an English editor goes through it to correct all the grammatical errors in existence throughout the manuscript. Line spacing should also be addressed throughout the manuscript.

[2] In this paper, the introduction reference to line 37-38 “Mercury, cadmium, lead, and arsenic are the most common heavy metal pollutants”. This part lacks the concentration limits of each metal to show which concentration of the metal is acceptable or tolerable, as recommended by WHO or other regulatory bodies. Other common heavy metals such as Chromium, Thallium and Antimony should be included in the above discussion.

[3] Line 48 “Traditional detection methods mainly calculate the concentration of an atom based on its characteristic spectral intensity, including atomic absorption spectroscopy, inductively coupled plasma mass spectroscopy, X-ray fluorescence spectrometry, neutron activation analysis and inductively coupled plasma-optical emission spectrometry”. The authors should include abbreviations to these analytical techniques.

[4] The authors should acquire copyright permission for all the figures, pictures, tables or schemes published elsewhere and add citations to all these figures.

[5] Figure 1, 3 and 4 should be referenced.

[6] “Table 1 summarizes the application of E-apt sensor in the detection of heavy metal ions in recent years”….There is no Table 1 in the manuscript.

[7] Line 285, “Its components, frequency-dependent resistors, and capacitance of the electrode are disturbed in AN AC mode”, what is an AN AC mode? Describe it in the main text.

[8] Line 357 “It also has some limitations, the main problem being the potential interference of electroactive compounds present in complex matrices”. This statement should be expanded further to explain how this problem is resolved.

[9] Line 410 “The forward and reverse TV pulse currents are recorded at the pulse time”, what does TV represent?

[10] The sensitivities of the sensors are often discussed in this manuscript, however, information on the selectivity of the same sensors is limited. It is critical that the sensors are both sensitive and selective, thus I urge the authors to add this information and describe how the aptamers influence the selectivity of the sensors.

[11] Future perspectives in terms of research opportunities and concepts requiring further research should be clearly deliberated in addition to the conclusion.

[12] The manuscript should be formatted according to the standard MDPI format, that includes author contributions, amongst others.

Reviewer 2 Report

This work reviews the use of nanomaterials to modify aptamer-based electro-chemical sensor developed for the detection of heavy metal ions.

 In general, the document is well organized, and well written. The manuscript requires minor revisions before being accepted, mainly related to the addition of some references. I also suggest revising English throughout the manuscript.

Being a review, I recommend deepening and increase the references to give more solidity to the manuscript. The suggestions are listed below:

  • line 70-71

“Aptamers have a wide range of recognition targets, such as proteins, small molecules, agricultural veterinary drugs, bacteria, and heavy metal ions, which have broad application prospects in the field of food detection” Here I would at least mention a couple of recently published works on aptasensors applied to the food sector

  • line 104

“In recent years, the successful synthesis of various nanomaterials has made them  widely used in chemical analysis, therapeutics, diagnostics, and food safety [15,16]” In addition to references 15 and 16 I suggest that you review the most recent bibliography and add some other examples.

  • line 295

“AuNPs are mainly used as electrodes and modification materials in EIS sensors” Here you could add references for of recently published works, to give greater truthfulness to the statement.

  • please remove section 5 “patents” if not relevant
